# Optimizing Treatment for Relapsed/Refractory Classic Hodgkin Lymphoma in the Era of Immunotherapy

**DOI:** 10.3390/cancers15184509

**Published:** 2023-09-11

**Authors:** Michael P. Randall, Michael A. Spinner

**Affiliations:** Division of Hematology/Oncology, Department of Medicine, University of California San Francisco, San Francisco, CA 94143, USA; michael.randall@ucsf.edu

**Keywords:** Hodgkin lymphoma, relapsed/refractory, brentuximab vedotin, checkpoint inhibitor, PD-1 inhibitor, nivolumab, pembrolizumab, autologous hematopoietic cell transplantation

## Abstract

**Simple Summary:**

Classic Hodgkin lymphoma (cHL) has a high cure rate with chemotherapy, but 10–30% of patients experience relapse or refractory (R/R) disease, depending on stage and risk factors. Treatment for R/R cHL differs between young, fit patients who are eligible for high-dose chemotherapy and autologous stem cell transplants and older adults who are not eligible for intensive therapies. Over the past decade, management of R/R cHL has evolved significantly following the approval of three highly active novel agents: brentuximab vedotin, nivolumab, and pembrolizumab, leading to improved cure rates and overall survival. In this review, we discuss our approach to the treatment of first relapse, maintenance therapy after transplant, relapse after transplant, and management of older adults and frail patients. Finally, we highlight emerging immunotherapies in clinical trials that hold great promise for the future.

**Abstract:**

Most patients with classic Hodgkin lymphoma (cHL) are cured with combination chemotherapy, but approximately 10–20% will relapse, and another 5–10% will have primary refractory disease. The treatment landscape of relapsed/refractory (R/R) cHL has evolved significantly over the past decade following the approval of brentuximab vedotin (BV), an anti-CD30 antibody-drug conjugate, and the PD-1 inhibitors nivolumab and pembrolizumab. These agents have significantly expanded options for salvage therapy prior to autologous hematopoietic cell transplantation (AHCT), post-transplant maintenance, and treatment of relapse after AHCT, which have led to improved survival in the modern era. In this review, we highlight our approach to the management of R/R cHL in 2023 with a focus on choosing first salvage therapy, post-transplant maintenance, and treatment of relapse after AHCT. We also discuss the management of older adults and transplant-ineligible patients, who require a separate approach. Finally, we review novel immunotherapy approaches in clinical trials, including combinations of PD-1 inhibitors with other immune-activating agents as well as novel antibody-drug conjugates, bispecific antibodies, and cellular immunotherapies. Ongoing studies assessing biomarkers of response to immunotherapy and dynamic biomarkers such as circulating tumor DNA may further inform treatment decisions and enable a more personalized approach in the future.

## 1. Introduction

Most patients with classic Hodgkin lymphoma (cHL) are cured with frontline combination chemotherapy; however, approximately 10–15% of patients with early-stage disease and 15–30% with advanced disease will relapse or have primary refractory disease [1,2,3,4]. Optimal treatment for relapsed or refractory (R/R) cHL differs between young, fit patients who are eligible for autologous hematopoietic cell transplantation (AHCT) and older adults or those with comorbidities who are ineligible for AHCT [5,6]. Over the past decade, the development of biologically targeted novel agents, brentuximab vedotin (BV) and programmed death-1 (PD-1) inhibitors, has reshaped the treatment landscape of R/R cHL and led to improved survival in the modern era [7]. In this review, we highlight recent studies integrating BV and PD-1 inhibitors into pre-transplant salvage therapy and post-transplant maintenance and discuss treatment options for post-transplant relapse with a focus on novel immunotherapy approaches in clinical trials. Based on the available evidence, we provide our current practice recommendations for the treatment of transplant-eligible and ineligible patients and the treatment of relapse after AHCT (Figure 1).

## 2. Management of Transplant Eligible Patients

For young, fit patients with R/R cHL, the current standard of care is salvage chemotherapy followed by AHCT [5,6]. Two randomized trials established AHCT as superior to chemotherapy alone, with significant improvements in progression-free survival (PFS) and over half of patients achieving a cure [8,9]. Many prognostic factors impact outcomes after AHCT, with primary refractory disease, early relapse within 1 year of initial therapy, and B symptoms or extra-nodal disease at relapse associated with poorer outcomes in the pre-PET era [10]. In the PET era, patients who undergo AHCT in metabolic complete remission (CR), defined in most studies as a Deauville score of 1–3, have superior outcomes [11]. In a retrospective study from Memorial Sloan Kettering of patients treated with platinum-based chemotherapy and AHCT, pre-transplant remission status by functional imaging (including gallium and FDG-PET) was the only statistically significant predictor of PFS and overall survival (OS) in multivariable analysis [12]. Patients achieving CR by functional imaging pre-transplant had a superior 5-year PFS of 75% versus 31% for patients with residual disease. Among patients with primary refractory disease, chemosensitivity to second-line therapy is a key prognostic factor for post-transplant outcomes, with a 10-year OS of 66% versus 17% in chemorefractory patients [13].

### 2.1. Pre-Transplant Salvage Therapy

#### 2.1.1. Combination Chemotherapy

Prior to the availability of biologically targeted novel agents, patients with R/R cHL after an anthracycline-based regimen were treated with alternative cytotoxic chemotherapy regimens prior to AHCT. In the second-line setting, platinum-based regimens including ifosfamide, carboplatin, and etoposide (ICE) [14], dexamethasone, cytarabine, and cisplatin (DHAP) [15], and etoposide, methylprednisolone, cytarabine, and cisplatin (ESHAP) [16] have overall response rates (ORR) ranging from 67–88% and CR rates ranging from 21–50% in the pre-PET era. Gemcitabine-based regimens including gemcitabine, dexamethasone, and cisplatin (GDP) [17], gemcitabine, vinorelbine, and liposomal doxorubicin (GVD) [18], and ifosfamide, gemcitabine, and vinorelbine (IGEV) [19] have similar activity with ORRs of 70–81% and CR rates of 19–54% in the pre-PET era. Bendamustine has also been combined with gemcitabine and vinorelbine (BEGEV) with a high CR rate of 75% in the PET era [20].

#### 2.1.2. BV-Based Regimens

Hodgkin and Reed-Sternberg (HRS) cells express CD30; however, their expression is limited in normal tissues to activated B and T lymphocytes, making it an ideal therapeutic target in cHL [21]. BV is a CD30-directed antibody-drug conjugate (ADC), which combines an anti-CD30 antibody with monomethyl auristatin E, a potent microtubule poison, via a protease- cleavable linker [22]. BV was initially approved by the United States Food and Drug Administration (FDA) in 2011 for the treatment of R/R cHL with progression after AHCT [23]. As a single agent in the multiply R/R setting, BV had an ORR of 75%, a CR rate of 34%, and a median PFS of 9.3 months, with durable responses observed in patients achieving CR [23,24]. The most notable adverse event (AE) was peripheral sensory neuropathy, which occurred in 42% of patients, including grade 3 neuropathy in 8%. Neuropathy was reversible in 90% of patients, with 73% experiencing complete resolution following treatment cessation [24].

In the pre-transplant salvage setting, BV has been evaluated as monotherapy, sequentially prior to chemotherapy, in combination with chemotherapy, and in combination with PD-1 inhibitors. In a phase 2 study in the pre-AHCT setting, single agent BV had an ORR of 75% and a CR rate of 43%, similar to the post-AHCT setting [25]. While the CR rate with single agent BV was notably lower than the PET CR rate observed with combination chemotherapy, this study provided proof of concept that a subset of patients could be spared chemotherapy prior to AHCT. Nearly half of the patients in this trial were able to proceed to AHCT without salvage chemotherapy, and no adverse impact on stem cell collection or engraftment was observed. Among patients proceeding directly to AHCT after BV, the 2-year PFS was 77%. Moskowitz et al. expanded upon this study with a PET-adapted protocol evaluating single agent BV followed by AHCT for patients achieving metabolic CR (27%), or 2 cycles of augmented ICE for patients with residual disease after BV [26]. 76% of patients achieved metabolic CR prior to AHCT using this strategy, and post-transplant outcomes were excellent with a 2-year PFS of 80%. Notably, there was no significant difference in outcomes between patients who were PET-negative after BV alone or after BV followed by augmented ICE [27]. Lynch et al. reported similar outcomes with concurrent dose-dense BV + ICE with a high PET CR rate of 74% in a cohort of patients enriched for primary refractory disease (64%) [28]. In addition to ICE, BV has been combined with other chemotherapy regimens, including DHAP, ESHAP, and bendamustine, with high PET CR rates of 70–79% and 2-year PFS rates of 70–76% post-transplant [29,30,31].

#### 2.1.3. PD-1 Inhibitor-Based Regimens

HRS cells have near universal genetic alterations of chromosome 9p24.1, leading to overexpression of the PD-1 ligand genes PD-L1 and PD-L2 [32]. In a seminal study by Roemer et al., 9p24.1 copy gain and amplification were present in 56% and 36% of cHL patients, respectively. 9p24.1 amplification correlated with greater PD-L1 expression by immunohistochemistry and was associated with advanced-stage disease and inferior PFS after frontline chemotherapy [33]. PD-L1/2 overexpression contributes to immune evasion by HRS cells and introduces a therapeutic vulnerability to checkpoint inhibitors targeting PD-1. In contrast to solid tumors, the likely mechanism of action of PD-1 blockade in cHL involves modulation of the tumor microenvironment (TME) with rapid clearance of CD4+ regulatory T cells and PD-L1+ macrophages and withdrawal of pro-survival signals rather than activation of CD8+ cytotoxic T cells [34,35,36]. Other recurrent molecular alterations, including NF-κB activation, PI3K signaling, microsatellite instability, high tumor mutational burden, and natural killer (NK) cell activation, may further contribute to the efficacy of PD-1 inhibitors in cHL [37,38].

Two PD-1 inhibitors, nivolumab and pembrolizumab, were FDA approved for R/R cHL in 2016 and 2017, respectively, based on the pivotal phase 2 CheckMate 205 and KEYNOTE-087 trials demonstrating high ORRs of 69–72%, CR rates of 16–27%, and median PFS of 14–15 months as single agents in the post-transplant setting [39,40,41]. In the pre-transplant setting, nivolumab and pembrolizumab have been combined with BV or cytotoxic chemotherapy with excellent results. In a phase 2 study of 93 patients enriched for primary refractory disease (42%) and early relapse after frontline therapy (30%), the combination of BV + nivolumab was highly active, with a PET CR rate of 67% and excellent 3-year PFS of 91% among patients proceeding directly to AHCT [42,43]. Excellent results have also been reported with nivolumab + ICE, pembrolizumab + ICE, and pembrolizumab + GVD, with remarkably high CR rates of 87–95% and 2-year PFS of 88–100% among patients proceeding to AHCT [44,45,46].

Historically, chemosensitivity has been considered a requirement for patients to benefit from AHCT; however, recent studies evaluating PD-1 inhibitors in the pre-transplant setting are challenging this paradigm. Several recent studies suggest that PD-1 inhibitors can sensitize cHL to subsequent cytotoxic chemotherapy, providing a biological rationale for earlier sequencing of these agents in the pre-transplant setting [47,48,49]. In one study of heavily pretreated patients with a median of 4 prior therapies, the ORR of chemotherapy administered after PD-1 blockade was 62%, exceeding that of prior chemotherapy regimens administered before PD-1 blockade [49]. Another study found that among patients previously refractory to chemotherapy who later received a PD-1 inhibitor, 82% were able to achieve a CR with chemotherapy, and 25 of 28 patients were able to proceed to subsequent allogeneic HCT [50]. In a multicenter retrospective study of 78 patients with multiply R/R cHL who underwent AHCT after a PD-1 inhibitor (54% refractory to 2 prior regimens), 18-month PFS was 81%, and response to PD-1 blockade was a better predictor of PFS than prior chemosensitivity [51].

With the increasing use of PD-1 inhibitor-based regimens in the pre-transplant setting, clinicians should be mindful of potential post-transplant toxicities that may be more common in this population. Recently, studies of patients undergoing AHCT after PD-1 blockade have identified a toxicity signal for engraftment syndrome, a potentially life-threatening complication characterized by fever, rash, diarrhea, elevated transaminases, and/or pulmonary infiltrates coinciding with neutrophil engraftment [52,53]. Retrospective studies have estimated rates of engraftment syndrome following PD-1 inhibitors as high as 77% [54], and the aforementioned phase 2 study of pembrolizumab + GVD reported engraftment syndrome in 68% of patients proceeding to AHCT [46]. In the phase 2 study of pembrolizumab + ICE, one patient experienced fatal acute respiratory failure attributed to peri-engraftment respiratory distress syndrome [45]. Of note, other studies of PD-1 inhibitor-based salvage regimens (e.g., nivolumab + ICE) have reported significantly lower rates of severe engraftment syndrome, which may be related in part to different definitions or criteria used. Further study is needed to determine the incidence and risk factors for engraftment syndrome; however, clinicians should remain vigilant for this potential toxicity in patients undergoing AHCT after PD-1 blockade, and early use of corticosteroids is recommended.

#### 2.1.4. Choosing Salvage Therapy Pre-Transplant

As summarized above and in Table 1, there are numerous options for pre-transplant salvage therapy in R/R cHL; however, prospective data comparing different regimens are currently lacking. A recent retrospective study from Stanford compared outcomes among 183 consecutive patients who underwent AHCT from 2011–2020 after receiving platinum-, gemcitabine-, BV-, or PD-1 inhibitor-based regimens. With a median follow-up of 4 years, PFS was significantly higher among patients receiving a PD-1 inhibitor compared to platinum-based chemotherapy (4-year PFS: 91% vs. 66%, *p* = 0.026) [7]. In multivariable analysis, receipt of a PD-1 inhibitor pre-transplant was associated with superior PFS with a hazard ratio of 0.21 (95% CI 0.05–0.80, *p* = 0.030). Similar results were published in a multicenter retrospective study of 936 patients from 14 centers in the U.S., which demonstrated superior PFS with PD-1 inhibitor-based regimens compared to other salvage therapies [55]. Comparing PD-1 inhibitor-based regimens, BV + chemotherapy, platinum-based chemotherapy, and BV alone, the 2-year PFS estimates after transplant were 79.7%, 62.3%, 49.6%, and 36.9%, respectively (*p* < 0.0001). In multivariable analysis, PD-1 inhibitor-based regimens pre-transplant were associated with superior PFS with a hazard ratio of 0.31 (95% CI 0.18–0.52, *p* < 0.0001) compared to platinum-based chemotherapy.

While these retrospective data are provocative, they should be considered hypothesis-generating, and a prospective randomized trial is needed to determine the optimal salvage regimen for R/R cHL. The phase 3 ECOG-ACRIN 4211 trial (EA4211) may answer this question by randomizing over 300 patients with R/R cHL after first-line therapy to receive the investigator’s choice of chemotherapy (ICE, GVD, or BV + bendamustine) with or without pembrolizumab (NCT05711628). Patients achieving CR or PR by PET will proceed to AHCT with the option to receive BV maintenance or radiotherapy (RT) consolidation. This study will open for accrual in 2023, and we encourage enrollment in this trial, which may ultimately change clinical practice.

When selecting initial salvage therapy for R/R cHL, we recommend a personalized approach considering the patient’s first-line treatment regimen, time to progression after initial therapy, and other patient and disease characteristics, including relevant comorbidities (e.g., peripheral neuropathy, autoimmune disorders), tumor bulk, and extent of disease. For all patients, we recommend early referral to an academic medical center experienced in AHCT and consideration of enrollment in the EA4211 trial. For patients with primary refractory disease or early relapse < 1 year after first-line therapy, we typically favor a regimen incorporating a PD-1 inhibitor in combination with non-cross-resistant chemotherapy, or BV [56].

### 2.2. Post-Transplant Maintenance/Consolidation

#### 2.2.1. BV Maintenance

Several trials have investigated whether post-transplant maintenance with BV and/or PD-1 inhibitors could improve outcomes for patients at high risk for relapse after AHCT. The phase 3 AETHERA trial randomized 329 patients with 1 or more high-risk factors (primary refractory disease, early relapse < 1 year after frontline therapy, or extranodal disease at relapse) to receive up to 16 cycles of BV or placebo every 3 weeks post-transplant. With mature follow-up, patients receiving BV maintenance had a superior 5-year PFS of 59% vs. 41% with placebo (HR 0.52, 95% CI 0.38–0.72), with greater benefit observed among patients with multiple risk factors for post-AHCT relapse [57]. Peripheral sensory neuropathy (56%) and neutropenia (35%) were the most common AEs. Notably, at 5-year follow-up, 90% of patients experienced resolution of neuropathy [57]. Of note, less than half of the patients in the BV group went on to complete all 16 intended cycles. However, retrospective data suggest that early discontinuation of BV does not compromise outcomes, with 2-year PFS of 89% vs. 86% for patients receiving >75% vs. 51–75% of the 16 cycles, respectively [58]. A recent retrospective study from 14 institutions in the U.S. and Czech Republic evaluated the impact of BV maintenance in a large real-world cohort [59]. Among 880 patients with high risk factors per the AETHERA trial, receipt of BV maintenance in 208 patients was associated with superior 2-year PFS (HR 0.7, 95% CI 0.5–0.9) and OS (HR 0.4, 95% CI 0.1–0.9). However, in propensity score weighted analysis, the PFS and OS benefits were observed only among patients with a PR by PET prior to AHCT. Based on the AETHERA trial and these real-world data, we offer BV maintenance to patients with one or more high-risk factors and strongly recommend BV maintenance for those with a PR prior to AHCT.

#### 2.2.2. PD-1 Inhibitor Maintenance

Both pembrolizumab and nivolumab have also demonstrated efficacy as post-transplant maintenance drugs. In a small phase 2 trial, 30 patients (90% with high risk factors per AETHERA) received up to 8 cycles of pembrolizumab as post-AHCT maintenance. After 18 months, PFS was 82% and OS was 100%. Among patients who would have been eligible for the AETHERA trial, 18-month PFS was 85% [60]. Similar outcomes have been reported with nivolumab monotherapy. Preliminary data from a trial of nivolumab maintenance reported a 6-month PFS of 92% and an OS of 100% [61]. Finally, Herrera et al. published a phase 2 trial evaluating the combination of BV + nivolumab as post-transplant maintenance. Fifty-nine patients (90% with primary refractory disease or early relapse) received 8 cycles of BV + nivolumab after AHCT. At 18 months, PFS was excellent at 94%. However, this regimen had significant toxicity, with 53% of patients developing peripheral neuropathy, 29% requiring corticosteroids for an immune-related AE, and 24% discontinuing treatment after a median of 4 cycles. Importantly, 51% of patients had prior exposure to BV and 42% had prior exposure to PD-1 blockade, suggesting that prior receipt of these therapies did not attenuate their efficacy in the post-transplant setting [62].

#### 2.2.3. Radiotherapy Consolidation

There is no randomized prospective trial evaluating the role of post-transplant radiotherapy (RT) consolidation in R/R cHL; however, several retrospective studies support this strategy, particularly for patients with early-stage disease and bulky sites. In a retrospective study of 64 patients who underwent AHCT at the University of Pennsylvania, receipt of consolidative RT in 17 patients was associated with improved local control at 3 years (78% vs. 48%, *p* = 0.02), defined as lack of recurrence at prior PET-positive sites [63]. In another retrospective study of 80 consecutive patients who underwent AHCT at the University of Minnesota, receipt of consolidative RT in 32 patients was associated with improved 2-year PFS of 67% vs. 42% (*p* < 0.01) with no difference in OS [64]. The PFS benefit remained significant in multivariable analysis (HR 4.64, 95% CI 1.98–10.88). In subgroup analysis, consolidative RT improved PFS in patients with bulky disease, B symptoms, primary refractory disease, and PR on pre-transplant PET. Based on these data, post-transplant RT consolidation is included as an option in the NCCN guidelines for R/R cHL [5]. In our practice, we consider RT consolidation for patients who are RT-naïve with localized bulky disease, particularly for those in PR pre-transplant and/or those who are poor candidates for BV maintenance.

### 2.3. Does Everyone with R/R cHL Need a Transplant?

The randomized trials that established AHCT as the standard of care for R/R cHL were notably conducted >20 years ago, preceding the development of BV and the PD-1 inhibitors [8,9]. With superb CR rates in the 70–95% range with novel salvage regimens in the second-line setting and highly effective maintenance strategies, there is growing interest in evaluating whether some patients may be spared the toxicity of AHCT or whether deferring transplant to later lines of therapy would affect outcomes [65]. A recent phase 2 trial evaluated the combination of tislelizumab (a novel PD-1 inhibitor) with gemcitabine and oxaliplatin (T-Gem/Ox) followed by 2 years of tislelizumab maintenance without AHCT in 30 patients with R/R cHL [66]. At a median follow-up of 16 months, 1-year PFS was 96% without AHCT. At least two other ongoing trials are evaluating novel salvage approaches that avoid AHCT. A phase 2 study at Memorial Sloan Kettering is evaluating salvage therapy with 4 cycles of pembrolizumab + GVD followed by 13 cycles of pembrolizumab maintenance without AHCT for patients achieving metabolic CR (NCT03618550). Another phase 2 study at City of Hope Medical Center is evaluating the combination of BV + nivolumab for 16 cycles in lieu of AHCT (NCT04561206). These approaches remain experimental; however, we encourage enrollment in these trials to answer these important questions about whether AHCT may be delayed or omitted in some patients.

## 3. Management of Transplant Ineligible Patients

Novel agents have significantly improved outcomes for R/R cHL patients who are unable to undergo AHCT due to advanced age, inadequate performance status, or comorbidities. As with transplant eligible patients, the choice of salvage therapy should be personalized, considering the first-line regimen, disease characteristics (tumor bulk, localized vs. systemic disease), and patient-related factors, including age, performance status, organ function, and comorbidities such as baseline neuropathy. For patients with localized recurrence, RT can be a highly effective definitive therapy and should be considered [67,68,69]. Clinical trials should also be prioritized in this population, including the aforementioned trials avoiding AHCT in CR2 (NCT03618550, NCT04561206).

For transplant-ineligible patients who have not previously received BV or a PD-1 inhibitor, we typically favor a regimen incorporating one or both of these agents. In the pivotal phase 2 trial of single agent BV, patients achieving CR (34%) had durable responses, with 38% of these patients remaining in remission for over 5 years [24]. Compared to single agent BV, pembrolizumab demonstrated a higher ORR (65.6% vs. 54.2%), a longer median PFS (13.2 vs. 8.3 months), and improved health-related quality of life in the randomized phase 3 KEYNOTE-204 trial, which was enriched for transplant-ineligible patients (63%) [70,71]. For patients responding to PD-1 inhibitors, we continue treatment for up to 1–2 years, depending on the depth of response. While the optimal duration of therapy is undefined, mature follow-up from the CheckMate 205 and KEYNOTE-087 trials established the feasibility of discontinuing nivolumab or pembrolizumab in patients achieving CR after 1 year of therapy [72,73]. While sample sizes are small, among patients who subsequently progressed after discontinuing immunotherapy, retreatment with nivolumab or pembrolizumab had similar response rates and PFS compared to the initial treatment course [74,75].

### Special Considerations with PD-1 Blockade

Because the immune response precipitated by PD-1 blockade can manifest as increased FDG avidity on PET, the Lugano criteria may not accurately capture responses. To account for this, a 2016 working group proposed the Lymphoma Response to Immunomodulatory Therapy Criteria (LYRIC) [76]. LYRIC refined the Lugano criteria to create a new “Indeterminate Response” category. Broadly, the IR category accounts for the early immune-mediated tumor flare (sometimes called “pseudo-progression”) and delayed responses characteristic of immune checkpoint blockade. While the LYRIC criteria have helped to harmonize response assessments, it remains important to consider the uncertainty introduced by immunomodulatory therapy when interpreting response rates in patients receiving these treatments. In addition to radiographic findings, clinicians should consider other clinical factors (e.g., B symptoms, pruritus) and laboratory parameters (e.g., ESR, hemoglobin, and albumin) when assessing response to immunotherapy.

There is also data to support continuing treatment with PD-1 blockade beyond disease progression, reflecting the uncertainty around how to optimally define progressive disease with these agents. Merryman et al. conducted a retrospective analysis of patients who received treatment with PD-1 inhibitors beyond progression (n = 20) compared to those who stopped PD-1 inhibitors at conventional progression on PET (n = 44). They found that the treatment beyond progression cohort had a longer PFS (17.5 vs. 6.1 months) and a longer time to subsequent treatment failure, despite being more heavily pre-treated [77]. This study supports continuing PD-1 inhibitors beyond radiographic disease progression in appropriate patients.

## 4. Treatment of Relapse after AHCT

Historically, patients who relapse after AHCT have had poor outcomes, with a median OS of just 2–3 years, and particularly poor outcomes for those with early relapse within 1 year after AHCT [78,79,80,81]. Recent studies of patients undergoing AHCT in the modern era demonstrate improved outcomes for this population, largely attributed to the approval of BV and the PD-1 inhibitors [7,82]. In a recent large retrospective study of 299 patients who relapsed after AHCT performed in 2011–2020, 5-year OS was 63%, and the median OS was 9.5 years after post-transplant progression [82]. OS was notably inferior among patients with early progression < 6 months after AHCT and among patients > 40 years old. OS was more favorable in patients receiving a PD-1 inhibitor-based regimen as their first post-transplant salvage compared to chemotherapy or BV-based regimens.

In general, treatment of relapse after AHCT varies based on patient-related factors (age, performance status, comorbidities), disease-related factors (tumor bulk, localized vs. systemic disease), and prior therapies. Most patients who relapse after AHCT are treated with palliative intent; however, a subset of patients may achieve cure or durable remissions. For patients with localized recurrence after AHCT, we consider post-transplant RT [67,68,69]. For patients with systemic relapse after AHCT, we favor a PD-1 inhibitor and/or BV-based regimen if not previously administered or if previously responsive to these agents. For young, fit patients, we recommend HLA typing to identify a suitable donor, and those who achieve CR or good PR with salvage therapy should be considered for consolidative allogeneic HCT.

A major unmet need is developing more effective therapies for so-called “triple-refractory” patients who have progressed after AHCT, BV, and a PD-1 inhibitor. For this very high-risk population, we recommend referral to an academic medical center and consideration of clinical trials, discussed at length in the final section of this review. Outside of a trial, treatment options are limited in this population and may include gemcitabine- or bendamustine-based regimens or oral agents such as everolimus or lenalidomide [5]. In a phase 2 study of multiply R/R cHL patients (75% with progression after AHCT), single agent bendamustine had an ORR of 53% and a CR rate of 33%, but with a short median response duration of 5 months [83]. Everolimus and lenalidomide have demonstrated modest single-agent activity in phase 2 trials in the post-AHCT setting with ORRs of 46% and 33%, respectively, and PRs in most responding patients [84,85]. A recent phase 1/2 trial combining temsirolimus with lenalidomide in 20 heavily pretreated patients (median 6 prior therapies) reported a more promising ORR of 80% and CR rate of 35%, allowing 40% of patients to proceed to allogeneic HCT [86]. These immunomodulatory agents may serve as a bridge to allogeneic HCT in select patients.

### Allogeneic HCT

In young, fit patients with multiply R/R cHL after AHCT and a suitable donor, allogeneic HCT represents a potentially curative treatment option; however, the potential risks of life-threatening graft-versus-host disease (GVHD) and infection remain important barriers to widespread adoption. Most studies have used reduced-intensity conditioning or nonmyeloablative regimens in this largely chemorefractory patient population [87,88,89]. In the era prior to post-transplant cyclophosphamide (PTCy) and PD-1 inhibitors, outcomes after allogeneic HCT were poor, with a 1-year cumulative incidence of relapse and non-relapse mortality (NRM) of 40% and 11%, respectively [90]. As with other hematologic malignancies, PTCy has been transformative in cHL, leading to a significant reduction in GVHD without compromising efficacy [91]. PD-1 blockade prior to allogeneic HCT also appears to enhance efficacy, with more favorable PFS, particularly among patients receiving haploidentical transplants and PTCy [92].

While allogeneic HCT following PD-1 blockade has been associated with an increased risk of severe acute GVHD, PTCy appears to mitigate this risk [93]. Merryman et al. published the largest retrospective cohort to date of 209 cHL patients from 33 centers in North America and Europe who underwent allogeneic HCT after a PD-1 inhibitor [89]. With a median follow-up of 24 months, the 2-year PFS and OS estimates were 69% and 82%, respectively, with grade 3–4 acute and chronic GVHD developing in 15% and 34% of patients, respectively. The 2-year cumulative incidence of relapse and NRM was 18% and 14%, respectively. In multivariable analysis, a longer interval from PD-1 blockade to allogeneic HCT and the use of PTCy as GVHD prophylaxis were associated with a reduced risk of GVHD and more favorable GVHD-free relapse-free survival [89]. Expanding on the role of PTCy, a single-center retrospective study from Johns Hopkins evaluated the outcomes of R/R cHL patients who received allogeneic HCT with PTCy, with the vast majority using haploidentical donors (85%) [94]. Comparing 37 patients who received PD-1 inhibitors prior to allogeneic HCT and 68 patients who did not, there was no significant difference in rates of acute or chronic GVHD. Post-transplant outcomes were excellent, with 3-year PFS and OS estimates of 90% and 94%, respectively, among patients treated with PD-1 inhibitors pre-transplant [94]. Based on the currently available evidence, we recommend a washout period of at least 6 weeks between PD-1 blockade and allogeneic HCT and the use of PTCy as GVHD prophylaxis [95].

## 5. Novel Immunotherapy Approaches

Immune evasion is a biological hallmark of cHL, with HRS cells evading an effective immune response through numerous mechanisms, including (1) expression of PD-1 ligands and other immune checkpoints on HRS cells and various immune cells in the TME; (2) downregulation of MHC molecules impairing antigen presentation to T cells; and (3) elaboration of cytokines and chemokines to maintain an immunosuppressive TME. The complex interaction between HRS cells and various immune cells in the TME is illustrated in Figure 2, which also highlights novel targets for immunotherapy. Completed and ongoing phase 1 and 2 trials of novel immunotherapy approaches in R/R cHL are discussed further below and summarized in Table 2.

### 5.1. PD-1 Inhibitor Combinations

Numerous immunotherapy trials seek to enhance the activity of PD-1 inhibitors by targeting additional checkpoint receptors on T cells (e.g., CTLA-4, LAG-3, TIM-3), adding immunomodulatory agents (e.g., lenalidomide, ibrutinib, ruxolitinib), or using epigenetic modifying therapies (e.g., hypomethylating agents, HDAC inhibitors) to uncover silenced checkpoints and overcome resistance to PD-1 blockade. Other trials have combined PD-1 inhibitors with antibodies that activate other immune effector cells, including macrophages (magrolimab) and NK cells (AFM13).

#### 5.1.1. PD-1 + CTLA-4 Blockade

The cHL TME is enriched for CLTA-4 + T cells, which are a distinct population from PD-1+ T cells and reside in close proximity to HRS cells [96]. HRS cells and some tumor-associated macrophages (TAMs) are also positive for CD86, the CTLA-4 ligand, suggesting a role for the CTLA-4/CD86 axis in contributing to immune evasion [96]. Several trials have combined nivolumab with the CTLA-4 inhibitor ipilimumab. The CheckMate 039 trial evaluated nivolumab + ipilimumab in 31 patients with R/R cHL, with an ORR of 74% and a CR rate of 23% [97]. Subsequently, a phase 1/2 ECOG trial evaluated BV in combination with nivolumab, ipilimumab, or both [98]. The CR rate was higher in the triplet arm compared to BV + nivolumab or BV + ipilimumab (73%, 61%, and 57%, respectively). However, the toxicity profile was also significantly greater in the triplet arm compared to BV + nivolumab, with grade 3–4 treatment-related AEs occurring in 50% vs. 16% of patients, including a higher incidence of rash and other immune-related AEs. Based on these results, a randomized phase 2 ECOG trial of BV + nivolumab with or without ipilimumab was recently conducted (NCT01896999). This study has completed accrual, and the results are eagerly awaited.

#### 5.1.2. PD-1 + LAG-3 or TIM-3 Blockade

As with other solid tumors, PD-1 inhibitors are being studied in combination with checkpoint inhibitors targeting LAG-3 and TIM-3, expressed on dysfunctional or exhausted T cells. Both LAG-3 and TIM-3 are expressed abundantly in the cHL TME, with ≥5% of T cells positive for these markers in 98% and 96% of cases, respectively [99]. HRS cells also express TIM-3 in approximately one-third of samples. A recent phase 1/2 study combined pembrolizumab with the anti-LAG3 antibody favezelimab in R/R cHL patients who were PD-1 naïve (cohort 1) or PD-1 refractory (cohort 2) [100]. In 34 patients treated in cohort 1, the ORR was 80%, the CR rate was 33%, and the median PFS was 19.4 months, which compares favorably to historical patients receiving pembrolizumab alone on the KEYNOTE-087 and KEYNOTE-204 trials. In 30 patients treated in cohort 2, the ORR was 29%, the CR rate was 9%, and the median PFS was 9.7 months. Some patients in both cohorts were successfully bridged to allogeneic HCT after study completion or discontinuation (17% and 6%, respectively). In animal models, combined blockade of PD-1 and TIM-3 with a bispecific antibody, AZD7789, inhibited tumor growth after progression on anti-PD-1 monotherapy [101]. A phase 1/2 study of AZD7789 is currently enrolling patients with R/R cHL after 2 or more therapies (NCT05216835).

#### 5.1.3. PD-1 + CELMoD

Lenalidomide is a cereblon E3 ligase modulator (CELMoD) that targets the critical B-cell transcription factors Ikaros and Aiolos for proteasomal degradation. In addition to its cytotoxic effects in B cells and plasma cells, lenalidomide has immunomodulatory properties, increasing interleukin-2 (IL-2) production, and may enhance T cell and NK cell responses when combined with PD-1 inhibitors [102]. Lenalidomide demonstrated modest single-agent activity in multiple R/R cHL patients in a phase 2 trial [85]. In a recent small phase 1b trial in 10 patients with R/R cHL (90% PD-1 naïve), the combination of nivolumab + lenalidomide had an ORR of 70% and a CR rate of 30%, and 4 responding patients were bridged to subsequent autologous or allogeneic HCT [103].

#### 5.1.4. PD-1 + BTK Inhibitor

Ibrutinib is a Bruton’s tyrosine kinase (BTK) inhibitor with immunomodulatory properties likely related to inhibition of IL-2 inducible T-cell kinase (ITK), which potentiates type 1 helper T-cell immune responses [104]. Ibrutinib has demonstrated single agent activity in small case series of heavily pretreated patients with cHL [105,106]. A phase 2 trial evaluated the combination of nivolumab + ibrutinib, hypothesizing that ibrutinib might promote a more favorable TME and enhance T-cell mediated immune responses. In 17 patients with a median of 5 prior therapies (59% PD-1 refractory), the ORR and CR rates were 52% and 29%, respectively with a median PFS of 17 months [107].

#### 5.1.5. PD-1 + JAK Inhibitor

Janus kinase 2 (JAK2) is overexpressed in HRS cells due to 9p24.1 copy gain and amplification and further increases transcription of the PD-1 ligand genes present at the same locus [32,33]. Constitutive activation of the JAK/STAT pathway is a biological hallmark of cHL, with frequent mutations in *STAT6*, *SOCS1*, *STAT3*, *STAT5B*, *JAK1*, *JAK2*, *PTPN1*, and other JAK/STAT pathway genes in HRS cells [108]. The JAK1/2 inhibitor ruxolitinib demonstrated a modest ORR of 19% in a phase 2 trial of heavily pretreated patients with R/R cHL [109]. The combination of ruxolitinib + nivolumab was more promising in a recent phase 1 trial in PD-1 refractory patients [110]. In 19 patients evaluable for response, the ORR was 42%, the CR rate was 26%, and the median response duration was 16.5 months. The selective JAK1 inhibitor itacitinib has also been studied in combination with everolimus in PD-1 refractory patients with promising preliminary results [111].

#### 5.1.6. PD-1 + Hypomethylating Agent

DNA methylation promotes T-cell exhaustion, and the hypomethylating agents (HMAs) azactidine and decitabine can reactivate T cells and overcome resistance to PD-1 blockade [112,113]. Low dose decitabine has been shown to increase tumor immunogenicity and may synergize with PD-1 inhibitors to restore immunosurveillance [114,115]. Several studies have evaluated HMA priming followed by PD-1 blockade with promising results. A phase 2 trial randomized PD-1 naïve patients 2:1 to receive decitabine priming for 5 days followed by camrelizumab (PD-1 inhibitor) or camrelizumab alone [116]. Patients previously treated with a PD-1 inhibitor were assigned to the combination arm. In PD-1 naïve patients, decitabine priming + camrelizumab led to a significantly higher CR rate compared to camrelizumab alone (71% vs. 32%, *p* = 0.003). In 50 patients with progression after a PD-1 inhibitor, decitabine priming + camrelizumab led to a promising ORR of 60%, a CR rate of 30%, and a median PFS of 21 months [117]. Phase 1 trials of oral azacitidine + nivolumab (NCT05162976) and oral decitabine/cedazuridine + nivolumab (NCT05272384) in PD-1 refractory patients are currently underway.

#### 5.1.7. PD-1 + HDAC Inhibitor

Histone deacetylase (HDAC) inhibitors have single agent activity in various lymphoma subtypes, modulate PD-1 expression, and have demonstrated synergy in combination with PD-1 inhibitors in preclinical studies [118]. A recent phase 1 study evaluated the HDAC inhibitor vorinostat in combination with pembrolizumab in 32 patients with R/R cHL, of whom 78% previously received a PD-1 inhibitor and 56% were PD-1 refractory [119]. For the entire cohort, the ORR and CR rates were 72% and 34%, respectively. Among PD-1 refractory patients, the ORR and CR rates were 56% and 11%, respectively. Grade 3 or higher AEs included hypertension (9%) and neutropenia (9%). Favorable results were also reported in a phase 2 trial of entinostat + pembrolizumab, with an ORR of 86% and CR rate of 45% in a small cohort of 22 patients (55% previously treated with a PD-1 inhibitor) [120]. In Epstein-Barr virus (EBV) positive cHL, HDAC inhibitors can sensitize lymphoma cells to nucleoside antiviral agents, and a phase 1b/2 trial of nanatinostat + valganciclovir has demonstrated activity in EBV+ lymphomas, including cHL [121].

#### 5.1.8. PD-1 + CD47 Blockade

CD47 is a “don’t eat me” signal expressed by numerous cancers to evade phagocytosis through interaction with its ligand SIRPα on TAMs and other phagocytes [122]. CD47 is overexpressed by HRS cells, and higher levels of CD47 expression correlate with inferior outcomes independent of PD-L1/2 expression [123,124]. TAMs are abundant in the cHL TME, and an increased number of TAMs is associated with inferior PFS, suggesting an important biological role in immune evasion [125,126]. Topological analysis of the cHL TME has identified PD-L1+ TAMs, monocytes, and dendritic cells, which physically co-localize with HRS cells and provide a biological rationale for combination checkpoint blockade targeting PD-1 and CD47 [127,128]. Magrolimab is an anti-CD47 antibody that promotes phagocytosis of lymphoma cells in preclinical models, polarizes macrophages from a tumor-tolerant M2 phenotype to an anti-tumor M1 phenotype, and has demonstrated synergy with rituximab in B-cell non-Hodgkin lymphomas [129,130]. Based on these preclinical data, an ongoing phase 2 trial is evaluating the combination of pembrolizumab + magrolimab in patients with R/R cHL after two or more therapies (NCT04788043).

#### 5.1.9. PD-1 + CD30/CD16A Bispecific Antibody

AFM13 is a novel CD30 × CD16A bispecific antibody that activates CD16A+ NK cells and macrophages to target CD30 + HRS cells [131,132]. AFM13 demonstrated modest single-agent activity in a phase 2 trial in multiply R/R cHL with an ORR of 17% [133]. More promising results were reported from a phase 1b study of 30 patients (all PD-1 naïve) treated with AFM13 + pembrolizumab [134]. At the highest dose level of AFM13 + pembrolizumab, the ORR and CR rates were 88% and 46%, respectively. The combination was well tolerated aside from a high incidence of infusion related reactions (IRR), including 13% of patients with grade 3 IRR with AFM13. As autologous NK cells are dysfunctional in lymphoma patients, a recent phase 1/2 trial combined AFM13 with allogeneic preactivated and expanded umbilical cord blood-derived NK cells in 30 patients with R/R CD30+ lymphomas (28 patients with cHL) [135]. Treatment was well tolerated, with no cases of GVHD, cytokine release syndrome (CRS), or immune effector cell-associated neurotoxicity syndrome (ICANS). Responses were excellent, with an ORR and CR rate of 97% and 63%, respectively, and five patients were bridged to consolidative HCT. At a median follow-up of 8 months, PFS and OS were 57% and 83%, respectively.

### 5.2. Novel Antibodies beyond PD-1

#### 5.2.1. Anti-CD25 ADC

Camidanlumab tesirine (cami-T) is an anti-CD25 antibody conjugated to a PBD dimer payload, which crosslinks DNA, leading to cell death [136]. Cami-T has two potential mechanisms of action in cHL: (1) direct cytotoxicity in CD25+ HRS cells (expressed in 60–80% of cases) and (2) depletion of immunosuppressive CD25+ regulatory T cells in the TME [137]. In a phase 1 study of 77 heavily pretreated patients (median 5 therapies, 74% with prior BV and PD-1), cami-T was highly active with an ORR of 71% and a CR rate of 42% [138]. Unfortunately, the toxicity profile was concerning, with 6% of patients developing Guillain-Barre syndrome (GBS). In a subsequent phase 2 trial in 117 patients (100% with prior BV and PD-1), cami-T demonstrated similar activity with an ORR of 70%, a CR rate of 33%, a median PFS of 9.1 months, and served as a bridge to autologous or allogeneic HCT in 14% of patients [139]. Unfortunately, 8 of 117 patients (6.8%) developed GBS or polyradiculopathy, with other notable toxicities including a maculopapular rash (33%), and edema or pleural effusions (17%). Most patients experiencing GBS had resolution or improvement of grade 1 symptoms following IVIG, corticosteroids, and/or plasmapheresis.

#### 5.2.2. Anti-TIGIT Antibody

T cell Ig and ITIM domains (TIGIT) are inhibitory immune checkpoint receptors that negatively regulate T cell and NK cell function. TIGIT is variably expressed by PD-1 + T cells in the cHL microenvironment, with higher levels of TIGIT expression in peritumoral lymphocytes correlating with lower levels of PD-L1 expression in HRS cells [140,141]. SEA-TGT is an anti-TIGIT monoclonal antibody associated with enhanced innate immune cell activation and augmented CD8 + T-cell responses in various solid tumors [142]. As part of a larger phase 1 trial (NCT04254107), SEA-TGT is being studied in combination with BV in multiple R/R cHL based on preclinical evidence of synergy.

### 5.3. Chimeric Antigen Receptor T-Cell Therapy

#### 5.3.1. Autologous CD30 CAR-T

While chimeric antigen receptor T-cell therapy (CAR-T) has revolutionized the treatment of R/R B-cell non-Hodgkin lymphomas, efficacy in cHL has been more limited. Most CAR-T constructs developed for cHL have targeted CD30 due to its highly specific expression on HRS cells. The phase 1/2 RELY-30 study enrolled 42 heavily pretreated patients (median 7 prior therapies, 100% with prior BV) [143]. Patients were treated at 3 dose levels following lymphodepletion with either fludarabine + cyclophosphamide or bendamustine with or without fludarabine. Treatment was well tolerated, with CRS occurring in 24% of patients (all grade 1) and no cases of ICANS. A characteristic rash occurred in 48% of patients but was self-limited. CD30 CAR-T was highly active, with an ORR of 72% and a CR of 59%. However, the durability of responses was disappointing, with a 1-year PFS of 36% and late relapses occurring beyond 1 year of follow-up. The phase 2 CHARIOT trial of CD30 CAR-T achieved a similar ORR of 73% in a heavily pre-treated patient population; however, the pivotal segment of this trial has been halted [144].

#### 5.3.2. Allogeneic CD30.CAR EBVSTs

Off-the-shelf, allogeneic anti-CD30 CAR-T products have also shown promise in R/R cHL. To avoid the risk of GVHD, one study modified EBV-specific T cells (EBVSTs) with a CAR targeting CD30 [145]. These CD30.CAR EBVSTs target CD30+ HRS cells and can be boosted in vivo in EBV+ recipients. In a phase 1 trial, 16 patients with heavily pretreated cHL (median of 5 prior therapies) received CD30.CAR EBVSTs were banked from 7 healthy donors at 4 escalating dose levels [146]. Treatment was well tolerated, with grade 1 CRS in 31% of patients and no cases of ICANS or GVHD. For all dose levels, the ORR and CR rates were 75% and 38%, respectively. However, the allogeneic T cells were very short lived with quantitative PCR for the CD30.CAR transgene showed near background levels within 1 week of infusion, suggesting elimination by alloreactive T cells. Of note, additional infusions of CD30.CAR EBVSTs from the same or different donor products led to second and third responses in some patients. Pre-clinical development of more advanced third generation CD30 CAR-T constructs continues with a focus on enhancing activity and durability [147].

**Table 2 cancers-15-04509-t002:** Novel immunotherapy approaches for relapsed/refractory classic Hodgkin lymphoma.

Regimen	Therapeutic Target(s)	Phase	Patient Population *	N	ORR	CRR	Median PFS	Median f/u	Ref
Nivo + ipilimumab	PD1 + CTLA4	1	4 prior lines	31	74%	23%	NR	18 mo.	[97]
Nivo + ipilimumab + BV	PD1 + CTLA4 + CD30	1/2	2 prior lines All PD1 naïve	64	82%	73%	NR	20 mo.	[98]
Pembro + favezelimab	PD1 + LAG3	1/2	PD1 naïve	30	80%	33%	19.4 mo.	32 mo.	[100]
Pembro + favezelimab	PD1 + LAG3	1/2	PD1 refractory	34	29%	9%	9.7 mo.	35 mo.	[100]
Nivo + lenalidomide	PD1 + CELMoD	1/2	3 prior lines 90% PD1 naïve	10	70%	30%	NR	--	[103]
Nivo + ibrutinib	PD1 + BTK/ITK	2	5 prior lines 59% prior PD1	17	52%	29%	17.3 mo.	9 mo.	[107]
Nivo + ruxolitinib	PD1 + JAK1/2	1/2	4 prior lines 100% prior PD1	21	42%	26%	16.5 mo.	21 mo.	[110]
Camrelizumab + decitabine	PD1 + HMA	2	PD1 naïve	61	95%	71%	NR	15 mo.	[116]
Camrelizumab + decitabine	PD1 + HMA	2	PD1 refractory	51	52%	36%	21.6 mo.	39 mo.	[117]
Pembro + vorinostat	PD1 + HDACi	1	3 prior lines 78% prior PD1	32	72%	34%	8.9 mo.	33 mo.	[119]
Pembro + entinostat	PD1 + HDACi	2	5 prior lines 55% prior PD1	22	86%	45%	NR	8 mo.	[120]
Pembro + AFM13	PD1 + CD30/CD16A	1	3 prior lines All PD1 naïve	30	83%	37%	--	--	[134]
AFM13 + umbilical cord blood derived NK cells	NK cells + CD30/CD16A	1/2	6 prior lines 29% prior PD1	30	97%	63%	NR	8 mo.	[135]
Camidanlumab tesirine	CD25	1	5 prior lines 74% prior PD1	77	71%	42%	6.8 mo.	9 mo.	[138]
Camidanlumab tesirine	CD25	2	5 prior lines 100% prior PD1	117	70%	33%	9.1 mo.	11 mo.	[139]
CD30 CAR-T (RELY-30)	CD30	1/2	7 prior lines	41	72%	59%	9 mo.	18 mo.	[143]
CD30 CAR-T (CHARIOT)	CD30	2	6 prior lines	15	73%	60%	--	--	[144]
CD30 CAR.EBVSTs	CD30 + EBV	1	5 prior lines	16	75%	38%	--	--	[146]

* prior lines indicate the median number of prior systemic therapies received prior to the study treatment. Legend: BTK, Bruton’s tyrosine kinase; BV, brentuximab vedotin; CAR-T, chimeric antigen receptor T-cell therapy; CELMoD, cereblon E3 ligase modulator; CRR, complete response rate; CTLA4, cytotoxic T-lymphocyte associated protein 4; EBV, Epstein-Barr virus; EBVSTs, Epstein-Barr virus-specific T cells; HDACi, histone deacetylase inhibitor; HMA, hypomethylating agent; ITK, interleukin-2 inducible T-cell kinase; JAK, Janus kinase; LAG3, lymphocyte-activation gene 3; N, number of patients; NK, natural killer; NR, not reached; ORR, overall response rate; PD1, programmed death-1; PFS, progression-free survival.

## 6. Conclusions and Future Directions

Treatment options for R/R cHL have expanded significantly over the past decade, now with numerous options for pre-transplant salvage therapy, post-transplant maintenance, and more effective regimens to treat post-transplant relapse and older adults ineligible for AHCT. The approval of BV and the PD-1 inhibitors has led to significant improvement in patient outcomes, with 4-year PFS/OS after AHCT increasing from 63%/79% in 2001–2010 to 73%/89% in 2011–2020 in a recent Stanford study [7]. Although initially approved in the post-transplant setting, BV and the PD-1 inhibitors are now increasingly used in earlier lines of therapy as part of the first salvage prior to AHCT or in the frontline setting for stage III-IV cHL based on the ECHELON-1 and SWOG S1826 trials [4,148,149]. The randomized phase 3 EA4211 trial will soon open for accrual and may help identify the optimal salvage regimen prior to AHCT. Several other trials will investigate whether AHCT may be deferred altogether in some patients and replaced with highly effective salvage regimens incorporating novel agents, followed by immunotherapy maintenance.

The remarkable efficacy of immunotherapy in cHL has shed light on important aspects of disease biology and the complex interplay between malignant HRS cells and the TME. These biological insights, along with new technologies such as single-cell RNA sequencing and mass cytometry, continue to enhance our understanding of cHL at the cellular and molecular level and will lay the foundation for the development of rational immunotherapy combinations and novel approaches to activate other immune effectors, including macrophages and NK cells. As in non-Hodgkin lymphomas, novel ADCs, bispecific antibodies, and CAR T-cell therapy have all demonstrated activity in cHL and hold promise for the future. With the increasing use of immunotherapy, it will be important to identify biomarkers that can predict response, such as 9p24.1 amplification, higher levels of PD-L1 expression, and MHC class II expression, all of which correlate with better responses to PD-1 blockade [150]. Finally, dynamic biomarkers such as circulating tumor DNA or metabolic tumor volume assessed longitudinally may further inform management decisions and enable a more personalized approach in the future [151,152,153,154].

## Figures and Tables

**Figure 1 cancers-15-04509-f001:**
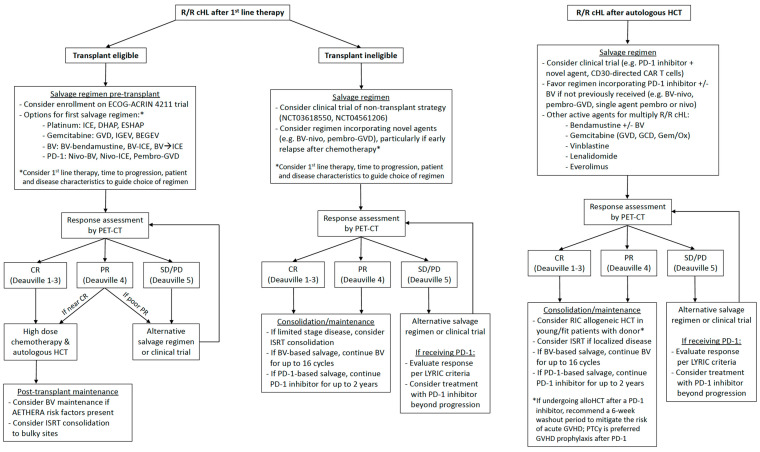
Suggested management algorithms for relapsed/refractory classic Hodgkin lymphoma.

**Figure 2 cancers-15-04509-f002:**
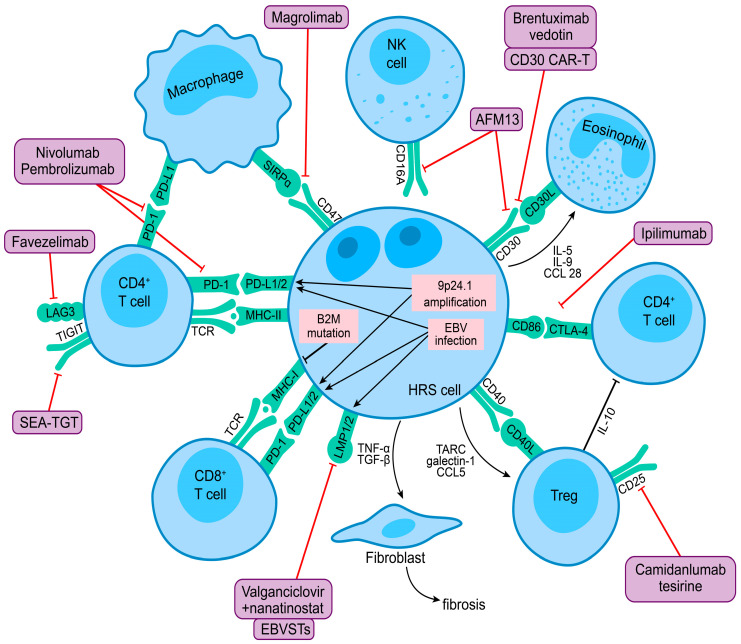
HRS cell interaction with the tumor microenvironment and novel targets for immunotherapy in the classic Hodgkin lymphoma illustration highlight the complex interplay between HRS cells and various immune cells in the tumor microenvironment. 9p24.1 Amplification and EBV infection in HRS cells lead to overexpression of PD-1 ligands, contributing to immune evasion. Frequent B2M mutations lead to loss of MHC class I expression and impaired antigen presentation to T cells. HRS cells produce cytokines and chemokines that recruit other immune cells and maintain an immunosuppressive microenvironment. Exhausted CD4 + T cells are abundant in the tumor microenvironment and express other checkpoint molecules, including CTLA-4, LAG-3, TIM-3, and TIGIT, which may prevent an effective immune response. CD47 expressed by HRS cells may impair phagocytosis through interaction with its ligand SIRPα on tumor-associated macrophages. The purple boxes show novel immunotherapies, and the red lines indicate their biological targets.

**Table 1 cancers-15-04509-t001:** Novel salvage regimens incorporating BV and/or PD-1 inhibitors prior to AHCT.

Regimen	Phase	N	CR Rate	PFS (All Patients)	PFS (AHCT Cohort)	Median Follow-Up	Reference
BV → augmented ICE	2	46	76%	82% (3 yrs)	82% (3 yrs)	20 mo.	[26]
BV + ICE	1/2	45	74%	80% (2 yrs)	NR	37 mo.	[28]
BV + DHAP	2	55	81%	74% (2 yrs)	NR	27 mo.	[29]
BV + ESHAP	1/2	66	70%	71% (2 yrs)	NR	27 mo.	[30]
BV + bendamustine	1/2	55	74%	63% (2 yrs)	70% (2 yrs)	21 mo.	[31]
Nivolumab + BV	1/2	93	67%	77% (3 yrs)	91% (3 yrs)	34 mo.	[43]
Nivolumab + ICE	2	37	91%	72% (2 yrs)	94% (2 yrs)	31 mo.	[44]
Pembrolizumab + ICE	2	42	87%	87% (2 yrs)	NR	24 mo.	[45]
Pembrolizumab + GVD	2	39	95%	100% (1 yr)	100% (1 yr)	14 mo.	[46]

Legend: AHCT, autologous hematopoietic cell transplantation; BV, brentuximab vedotin; CR, complete response; DHAP, dexamethasone, cytarabine, cisplatin; ESHAP, etoposide, methylprednisolone, cytarabine, cisplatin; GVD, gemcitabine, vinorelbine, liposomal doxorubicin; ICE, ifosfamide, carboplatin, etoposide; N, number of patients; NR, not reported; PFS, progression-free survival.

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
