# Peer review of "Optimizing Treatment for Relapsed/Refractory Classic Hodgkin Lymphoma in the Era of Immunotherapy"

_cancers, 2023, doi:10.3390/cancers15184509_

Round 1

Reviewer 1 Report

In this research, the authors reviewed the recent development of Optimizing treatment for relapsed/refractory classic Hodgkin lymphoma in the era of immunotherapy. Generally, it’s meaningful and interesting review. In my opinion, the current version of this manuscript fits the scope of Cancers and could be accepted after major revision.

My specific comments are in detail listed below:

1.  The effects of immunotherapy on T cells, DC function, and macrophage polarization could be also revealed in this review if possible.

2.     In this review “Novel immunotherapy approaches”, how PD-L1 targeting strategy was used to treat the relapsed/refractory classic Hodgkin lymphoma or some related cancers should be added. Some references could be added including 10.1002/adma.202206121.

3.     A better conclusion that conclude or predict how immunotherapy affect the therapy status of relapsed/refractory classic Hodgkin lymphoma should more deeply discussed.

4.     How PD-L1 targeting strategy affect the era of immunotherapy in the treatment for relapsed/refractory classic Hodgkin lymphoma bladder cancers or some other tumors should be added to this review. Some references should be added to this part including 10.1016/j.ijbiomac.2022.10.167.

5.     Some minor mistakes existed in the references. The authors should carefully check it.

Author Response

September 3, 2023

We thank the reviewers for their helpful comments and review of our manuscript. We have responded to all reviewer comments below in red text and have attached an updated version of the manuscript with our edits highlighted in red text.

This revision has further strengthened the manuscript, which we believe will be of great interest to the readership of Cancers. We thank the reviewers for the time and effort spent in the review of our paper.

Sincerely,

Michael P. Randall, MD and Michael A. Spinner, MD

Reviewer #1 Comments

  1. The effects of immunotherapy on T cells, DC function, and macrophage polarization could be also revealed in this review if possible.

We highlight the various mechanisms of action of the PD-1 inhibitors in cHL in lines 106-116, including their effects on T cells, NK cells, and macrophages. We have expanded our discussion of tumor-associated macrophages and other mononuclear phagocytes including DCs in the cHL tumor microenvironment (see lines 432-442). We also discuss macrophage checkpoint inhibitors, which have been shown to polarize macrophages from a tumor-tolerant M2 phenotype to an anti-tumor M1 phenotype. Figure 2 shows the interaction between HRS cells and the immune microenvironment, including CD4+ T cells, CD8+ T cells, CD25+ Tregs, and tumor-associated macrophages.

  1. In this review “Novel immunotherapy approaches”, how PD-L1 targeting strategy was used to treat the relapsed/refractory classic Hodgkin lymphoma or some related cancers should be added. Some references could be added including 10.1002/adma.202206121.

PD-L1 inhibitors like avelumab have been studied in R/R cHL, for example in the phase 1b JAVELIN trial (PMID 34477818). However, the ORR and CR rate appear lower with agents targeting PD-L1 (the ligand) compared to agents targeting PD-1 (the receptor) in R/R cHL. Ongoing immunotherapy trials in R/R cHL are thus more focused on building upon a PD-1 inhibitor backbone, and we have highlighted the key trials building upon nivolumab or pembrolizumab in this review article and in Table 2. We also briefly discuss trials incorporating other PD-1 inhibitors including tislelizumab and camrelizumab. For the sake of brevity and clarity, we have focused our review article on cHL rather than expanding the manuscript to cover other tumor types.

  1. A better conclusion that conclude or predict how immunotherapy affect the therapy status of relapsed/refractory classic Hodgkin lymphoma should more deeply discussed.

In the conclusion, we discuss how approval of the PD-1 inhibitors has translated to significant improvements in PFS and OS for patients with R/R cHL. We discuss how the PD-1 inhibitors are moving into earlier lines of therapy, including as first salvage therapy prior to autoHCT, and in the frontline setting for patients with stage III-IV cHL based on the SWOG S1826 trial. We also discuss ongoing trials evaluating PD-1 inhibitor-based salvage regimens followed by PD-1 inhibitor maintenance in an effort to avoid autoHCT altogether (discussed in more detail in Section 2.3). Finally, we discuss how a major effort of ongoing trials is to build upon PD-1 inhibitors by adding other checkpoint inhibitors or immunomodulatory agents, and discuss other novel immunotherapeutic agents like ADCs, bispecific antibodies, and CAR T-cell therapy. A practical approach to how the authors use PD-1 inhibitors for R/R cHL in clinical practice is provided in Figure 1.

  1. How PD-L1 targeting strategy affect the era of immunotherapy in the treatment for relapsed/refractory classic Hodgkin lymphoma bladder cancers or some other tumors should be added to this review. Some references should be added to this part including 10.1016/j.ijbiomac.2022.10.167.

This review article focuses specifically on immunotherapy for R/R cHL. As discussed in our review, the mechanism of action of PD-1 inhibitors in cHL differs from that of solid tumors (see lines 106-116), and relates to genetic alterations of chromosome 9p24.1, a defining feature in HRS cells. The role of PD-1/PD-L1 inhibitors in bladder cancer is outside the scope of this manuscript and is reviewed elsewhere (see PMID 30275703, 28214651).

  1. Some minor mistakes existed in the references. The authors should carefully check it.

We have reviewed and updated the references.

Reviewer #2 comments

The authors presented in a very structured way all the therapeutic solutions for patients with refractory or relapsed Hodgkin's lymphoma, making relevant considerations on the therapeutic results. The newest therapeutic approaches in the field are presented - CART, checkpoint inhibitors, monoclonal antibodies, hypomethylating agents, etc. in addition to the classic approach represented by autologous or allogeneic transplantation. The review is very well organized and meets all the conditions to be accepted for publication.

We thank the reviewer for the positive feedback and review of our manuscript.

Reviewer #3 comments

Dear authors,

Thank you for the opportunity to review this manuscript.  I really enjoyed reading it, and overall I find it to be very well written.  My suggestions:

2.1.3 Lines 100-110 - Please use the Herrera, et al. Ann Onc 2018 reference for the COH trial and not the Chen, et al BBMT 2015 since the former includes all of the cohorts and longer post-HCT follow-up.  Overall CR was 43% for 2L there.  Notably, COH defined CR as DS 1-3 whereas the MSKCC BV -> augICE defined as DS1-2 which might account for the difference in CR rates.  2y post-HCT PFS was 77% for COH pts proceeding directly to ASCT off of BV.

We thank the reviewer for this helpful comment. We have updated the text and reference (see lines 89-90, 94-95, and new reference 25).

LInes 161-173 - may be worth mentioning that data are somewhat variable re: engraftment syndrome between studies.  Nivo-ICe with relatively low rates of severe ES.

We have added a statement on the disparate rates of engraftment syndrome reported across trials, perhaps related in part to different definitions or criteria used (see lines 145-149).

2.1.4.  Lines 206-208 - Obviously this is physician / institution preference with no answer, but just to point out that BV/nivo had a pretty low rate of CR in the primary refractory subset.  As such, for early relapse (and especially primary refractory), my strong preference is PD-1 + chemo.  However, this is my personal opinion only with the usual caveats of drawing too much significance from a subset analysis.

We thank the reviewer for this comment. For patients with early relapse or primary refractory disease, our preference is for a PD-1 inhibitor-based regimen, typically pembro-GVD or BV/nivo, also depending in part on the frontline regimen and disease burden. We cite an ASCO abstract that reported improved PFS with BV/nivo compared to other salvage regimens (e.g. BV/benda, platinum, and gemcitabine-based regimens) in a large multicenter retrospective cohort of R/R cHL patients with early relapse or primary refractory disease (see lines 176-178 and reference 56).

2.2.1 BV maintenance - please cite Wagner CB, et al. Haematologica 2023 which supports the notion that maybe ending BV before 16 doses isn't that big a deal from an efficacy standpoint.

We added this reference and a statement in the text to convey this point (see lines 190-191).

2.2.2 Please mention toxicities of BV/nivo maintenance - they were substantial!

We have added a summary of the major toxicities of BV/nivo maintenance in the text (see lines 207-209).

  1. (lines 305-308) please provide citations for retreatment efficacy of nivo/pembro

We have added the following citations for retreatment efficacy with nivo or pembro: Manson et al, Haematologica 2020, PMID: 33131257, Fedorova et al, Ann Hematol 2021, PMID: 33528609. See lines 258-260 and references 74-75.

  1. (lines 361-363) would cite Major A, et al. BJH 2022 re: lenalidomide / temsirolimus as bridge to allo as another option -- small numbers but the ORR in HL is very compelling.

We thank the reviewer for this comment and reference. We have added this citation and discuss this combination in the text (see lines 305-307 and reference 86).

4.1. This is always a challenging discussion (what to expect from allo these days).  I do think it's helpful to mention that historically allo for HL was not a great proposition - relapse rate around 40% with high rates of GVHD / NRM made it so that few pts enjoyed ongoing GRFS.  However, PTCy has been transformative and relapse rates are clearly lower than before with much less GVHD.  I think that's the overall framework, and it would be helpful to include that outcomes with allo are significant better, likely in large part due to the use of PTCy which was decreased GVHD significantly as it does for all allo without sacrificing efficacy (in fact, relapse seems lower than before too -- this may be driven by increased use of PD-1 blockade prior to allo which really seems to decrease relapse -- the EBMT had a nice paper about this from De Philippis et al Blood Advances 2020).

We thank the reviewer for this helpful comment and reference. We have added the De Phillippis reference and revised our discussion of allogeneic HCT to more clearly highlight how PTCy and PD-1 inhibitors have changed post-transplant outcomes for R/R cHL (see lines 313-320 and new references 90-92).

5.3.1. CHARIOT is unfortunately finished as Tessa therapeutics has gone bankrupt.  I would mention that the trial has been halted.

Thank you for alerting us. We now mention this in the text (see line 491).

Reviewer 2 Report

The authors presented in a very structured way all the therapeutic solutions for patients with refractory or relapsed Hodgkin's lymphoma, making relevant considerations on the therapeutic results. The newest therapeutic approaches in the field are presented - CART, checkpoint inhibitors, monoclonal antibodies, hypomethylating agents, etc. in addition to the classic approach represented by autologous or allogeneic transplantation. The review is very well organized and meets all the conditions to be accepted for publication.

Author Response

(The authors gave the same response as above.)

Reviewer 3 Report

Dear authors,

Thank you for the opportunity to review this manuscript.  I really enjoyed reading it, and overall I find it to be very well written.  My suggestions:

2.1.3 Lines 100-110 - Please use the Herrera, et al. Ann Onc 2018 reference for the COH trial and not the Chen, et al BBMT 2015 since the former includes all of the cohorts and longer post-HCT follow-up.  Overall CR was 43% for 2L there.  Notably, COH defined CR as DS 1-3 whereas the MSKCC BV -> augICE defined as DS1-2 which might account for the difference in CR rates.  2y post-HCT PFS was 77% for COH pts proceeding directly to ASCT off of BV.

LInes 161-173 - may be worth mentioning that data are somewhat variable re: engraftment syndrome between studies.  Nivo-ICe with relatively low rates of severe ES.

2.1.4.  Lines 206-208 - Obviously this is physician / institution preference with no answer, but just to point out that BV/nivo had a pretty low rate of CR in the primary refractory subset.  As such, for early relapse (and especially primary refractory), my strong preference is PD-1 + chemo.  However, this is my personal opinion only with the usual caveats of drawing too much significance from a subset analysis.

2.2.1 BV maintenance - please cite Wagner CB, et al. Haematologica 2023 which supports the notion that maybe ending BV before 16 doses isn't that big a deal from an efficacy standpoint.

2.2.2 Please mention toxicities of BV/nivo maintenance - they were substantial!

3. (lines 305-308) please provide citations for retreatment efficacy of nivo/pembro

4. (lines 361-363) would cite Major A, et al. BJH 2022 re: lenalidomide / temsirolimus as bridge to allo as another option -- small numbers but the ORR in HL is very compelling.

4.1. This is always a challenging discussion (what to expect from allo these days).  I do think it's helpful to mention that historically allo for HL was not a great proposition - relapse rate around 40% with high rates of GVHD / NRM made it so that few pts enjoyed ongoing GRFS.  However, PTCy has been transformative and relapse rates are clearly lower than before with much less GVHD.  I think that's the overall framework, and it would be helpful to include that outcomes with allo are significant better, likely in large part due to the use of PTCy which was decreased GVHD significantly as it does for all allo without sacrificing efficacy (in fact, relapse seems lower than before too -- this may be driven by increased use of PD-1 blockade prior to allo which really seems to decrease relapse -- the EBMT had a nice paper about this from De Philippis et al Blood Advances 2020).

5.3.1. CHARIOT is unfortunately finished as Tessa therapeutics has gone bankrupt.  I would mention that the trial has been halted.

Author Response

(The authors gave the same response as above.)

Round 2

Reviewer 1 Report

The current version of this manuscript could be accepted.